# Added value of clinical prediction rules for bacteremia in hemodialysis patients: An external validation study

Sho Sasaki[1,2,3☯]*, Yoshihiko Raita[4,5☯], Minoru Murakami[6], Shungo Yamamoto[3,7], Kentaro Tochitani[3,7], Takeshi Hasegawa[8,9,10], Kiichiro Fujisaki[1], Shunichi Fukuhara[10,11,12]

**1** Department of Nephrology, Iizuka Hospital, Fukuoka, Japan, **2** Clinical Research Support Office, Iizuka Hospital, Fukuoka, Japan, **3** Department of Healthcare Epidemiology, Kyoto University Graduate School of Public Health, Kyoto, Japan, **4** Department of Nephrology, Okinawa Prefectural Chubu Hospital, Naha, Japan, **5** Department of Emergency Medicine, Massachusetts General Hospital, Harvard Medical School, Boston, MA, United States of America, **6** Department of Nephrology, Saku Central Hospital, Nagano, Japan, **7** Department of Infectious Disease, Kyoto City Hospital, Kyoto, Japan, **8** Office for Promoting Medical Research, Showa University, Tokyo, Japan, **9** Division of Nephrology, Department of Medicine, Showa University Fujigaoka Hospital, Yokohama, Japan, **10** Fukushima Medical University, Fukushima, Japan, **11** Section of Clinical Epidemiology, Department of Community Medicine, Kyoto University, Kyoto, Japan, **12** Department of Health Policy and Management, Johns Hopkins Bloomberg School of Public Health, Baltimore, MD, United States of America

☯ These authors contributed equally to this work.
* ssasakih4@aih-net.com

**Data Availability Statement:** All relevant data are within the manuscript and its Supporting Information files.

## Abstract

### Introduction

Having developed a clinical prediction rule (CPR) for bacteremia among hemodialysis (HD) outpatients (BAC-HD score), we performed external validation.

### Materials & methods

Data were collected on maintenance HD patients at two Japanese tertiary-care hospitals from January 2013 to December 2015. We enrolled 429 consecutive patients (aged ≥ 18 y) on maintenance HD who had had two sets of blood cultures drawn on admission to assess for bacteremia. We validated the predictive ability of the CPR using two validation cohorts. Index tests were the BAC-HD score and a CPR developed by Shapiro et al. The outcome was bacteremia, based on the results of the admission blood cultures. For added value, we also measured changes in the area under the receiver operating characteristic curve (AUC) using logistic regression and Net Reclassification Improvement (NRI), in which each CPR was added to the basic model.

### Results

In Validation cohort 1 (360 subjects), compared to a Model 1 (Basic Model) AUC of 0.69 (95% confidence interval [95% CI]: 0.59–0.80), the AUC of Model 2 (Basic model + BAC-HD score) and Model 3 (Basic model + Shapiro's score) increased to 0.8 (95% CI: 0.71–0.88)

**Funding:** No, the authors have not received a specific grant for this study from any funding agency in the public, commercial, or not-for-profit sectors.

**Competing interests:** NO authors have competing interests

and 0.73 (95% CI: 0.63–0.83), respectively. In validation cohort 2 (96 subjects), compared to a Model 1 AUC of 0.81 (95% CI: 0.68–0.94), the AUCs of Model 2 and Model 3 increased to 0.83 (95% CI: 0.72–0.95) and 0.85 (95% CI: 0.76–0.94), respectively. NRIs on addition of the BAC-HD score and Shapiro's score were 0.3 and 0.06 in Validation cohort 1, and 0.27 and 0.13, respectively, in Validation cohort 2.

## Conclusion

Either the BAC-HD score or Shapiro's score may improve the ability to diagnose bacteremia in HD patients. Reclassification was better with the BAC-HD score.

## Introduction

Bacteremia is a disease with a high mortality rate [1–4]. Early diagnosis and treatment are keys to improving prognosis. However, due to the variety of clinical presentations of bacteremia, it is not always first on the list of possible diagnoses. In this context, several clinical predictive models (CPRs) of bacteremia in the general population have been developed [5–8]. Among these models, those developed by Shapiro et al. (Shapiro's model) [9] have been widely validated and internationally recognized [10–12].

Patients on hemodialysis (HD) are known to have a higher morbidity and mortality rate from bacteremia compared to the general population. Previous cohort studies have shown that the incidence of bacteremia in patients on maintenance HD is 10.40–18.98 per 100 person-years [13–16], which is much higher than the incidence in the general population of 0.22 per 100 person-years [1]. The annual mortality due to sepsis, a severe complication of bacteremia, in HD patients is 50–100 times higher than that of the general population [17, 18].

The most frequent cause of bacteremia in the general population is urinary tract infections caused by *Escherichia coli* [19–22], whereas the most frequent cause in HD is *Staphylococcus aureus* [23–25]. Since many blood test findings in HD patients are often affected by dialysis, the accuracy of items included in existing CPRs developed in the general population, such as blood counts and serum creatinine levels, may be greatly affected.

[12]. In order to address the peculiarities of bacteremia among patients with HD, we developed a CPR specific to bacteremia in HD patients, the BAC-HD score [26].

Since the external validities of Shapiro's model [12] and the BAC-HD score in patients on maintenance HD have not been verified, we assessed the diagnostic accuracy of these two prediction models for bacteremia.

## Materials and methods

The present study was approved by the ethics committees of Iizuka Hospital (17167–436), Okinawa Prefectural Chubu Hospital (H28-51) and Saku General Hospital (R201701-01). The study was conducted in accordance with the ethical standards of the Declaration of Helsinki. Since all patient information analyzed in this study was retrospective, participants' written informed consent was not required by the ethics committee. All data were fully anonymized before authors accessed them. We accessed the medical records to obtain data at Okinawa Prefectural Chubu Hospital from February 9th to February 13th, 2017, and at Saku General Hospital from February 24th to 26th, 2017. The study results are reported in accordance with the Standards for Reporting Diagnostic Accuracy (STARD) statement [27].

## Study design and participants

We conducted a cross-sectional study of maintenance HD patients at two tertiary-care teaching hospitals.

Data were collected from medical records from January 2013 to December 2015 in each facility. We enrolled consecutive participants on maintenance HD who were aged $\geq$ 18 y with two sets of blood cultures drawn at admission because of suspicion of bacteremia. Exclusion criteria were participants who met any of the following items: 1) inpatients transferred from another hospital, 2) patients with a vintage of dialysis < 2 months, 3) patients also receiving peritoneal dialysis, and 4) patients receiving HD less than once a week.

## Index tests

Two clinical prediction rules (CPR) for bacteremia were adopted as index tests.

The first CPR, the BAC-HD score, consists of the following 5 items: 1) body temperature $\geq$ 38.3˚C, 2) heart rate $\geq$ 125, 3) C-reactive protein $\geq 10 \times 10^4$μg/L, 4) alkaline phosphatase > 6 μkat IU/L, and 5) no prior antibiotic use within the past week. Each item is regarded as 1 point, for a maximum total of 5 points.

Shapiro's score is composed of "major criteria" defined as: 1) temperature > 39.5˚C (3 points), 2) indwelling vascular catheter (2 points) or clinical suspicion of endocarditis (3 points); plus "minor criteria" (1 point each) defined as: 1) temperature 38.3–39.4˚C (101–102.9˚F), 2) age > 65 y, 3) chills, 4) vomiting, 5) hypotension (systolic blood pressure < 90 mmHg), 6) white blood cell count $> 18 \times 10^9$/L, 7) bands > 5%, 8) platelets $< 150 \times 10^9$/L, and 9) creatinine > 176.8 μmol/L.

## Reference standard

The reference standard was bacteremia, as per the results of the admission blood cultures. Contamination was defined as: one of the two sets of culture bottles was positive, or cases where certain species of bacteria known to be contaminants, such as diphtheroids, *Bacillus spp*., *Propionibacterium spp*., *Micrococci*, *Corynebacterium spp*., and coagulase-negative *Staphylococci* were detected. Finally, an external consensus panel of infectious disease physicians with > 10 y clinical experience and Japanese Board of Infectious Disease certification who were blinded to the present study design determined whether a culture was contaminated or not based on the above definitions and their clinical expertise.

## Statistical analysis

**Validation cohorts.** Since there are no standard criteria for obtaining blood cultures, it was suspected that the selection of subjects may have differed depending on the facility. Therefore, two validation cohorts were set based on the logic that it is desirable to verify the validity at multiple facilities. Validation cohort 1 (360 subjects) and validation cohort 2 (96 subjects) were defined as patients at Okinawa Prefectural Chubu Hospital and Saku General Hospital, respectively.

**Descriptive statistics.** We analyzed each item with the two CPRs and used proven bacteremia as a reference standard as well as other clinical information, including sex, blood pressure, respiratory rate, hemodialysis vintage, and presence of diabetes mellitus. Continuous and categorical variables are presented as the median (interquartile range: IQR) and number (percentage), respectively (Table 1).

**Basic model.** The basic model to assess the value of reclassification of CPRs was conducted using a logistic regression model with explanatory variables of sex (0: female, 1: male),

**Table 1. Baseline characteristics.**

| | Validation cohort 1 | | | Validation cohort 2 | | |
|---|---|---|---|---|---|---|
| | *n* = 360 | | | *n* = 96 | | |
| | Bacteremia (-) | Bacteremia (+) | Missing (*n*) | Bacteremia (-) | Bacteremia (+) | Missing (*n*) |
| | *n* = 323 | *n* = 37 | | *n* = 80 | *n* = 16 | |
| Age (years), median (IQR) | 72 (61–79) | 73 (62–79) | 0 | 72 (65–81) | 71.5 (68.5–82.5) | 0 |
| sex | | | 0 | | | |
| male | 185 (42.7) | 16 (43.2) | | 56 (70) | 12 (75) | |
| female | 138 (57.3) | 21 (56.8) | | 24 (30) | 4 (25) | |
| vital signs, median (IQR) | | | | | | |
| systolic blood pressure (mmHg) | 137 (110–153) | 140 (109–150) | 12 | 141 (129–160) | 134 (118–155) | 2 |
| diastolic blood pressure (mmHg) | 70 (60–80) | 64 (60–70) | 36 | 76 (63–87) | 72 (57–84) | 2 |
| pulse rate (/min) | 88 (78–100) | 93.5 (83–110) | 17 | 83 (72–94) | 86 (74–95) | 5 |
| respiratory rate (/min) | 20 (18–24) | 20 (18–24) | 22 | 20 (16–25) | 18 (16–24) | 76 |
| body temperature (˚C) | 37.1 (36.5–37.7) | 37.6 (36.7–38.9) | 12 | 37.4 (37–38) | 38.1 (37.8–38.8) | 17 |
| laboratory findings, median (IQR) | | | | | | |
| white blood cell count ($\times 10^9$/L) | 8.1 (6–11.3) | 9.9 (5.9–13.2) | 5 | 7.5 (5.4–9.4) | 8.1 (5.9–14) | 14 |
| platelet count ($\times 10^9$/L) | 168 (128–219) | 137 (109–195) | 6 | 148 (116–189) | 89 (70–168) | 14 |
| alkaline phosphatase (μkat/L) | 4.6 (3.7–5.9) | 6.1 (4.0–8.2) | 182 | 4.3 (3.5–5.4) | 5.1 (3.2–8.1) | 19 |
| creatinine (μg/L) | 539.2 (389.0–751.4) | 495.0 (371.3–654.2) | 5 | 610.0 (503.9–751.4) | 539.2 (415.5–830.96) | 23 |
| C-reactive protein ($\times 10^4$μmol/L) | 4.2 (1.5–9.9) | 6.7 (1.6–15) | 45 | 4.2 (1.3–10.1) | 6.0 (2.6–11.6) | 17 |
| hemodialysis vintage (months), median (IQR) | 48.8 (16.6–116) | 49 (18.7–92.8) | 17 | 87 (44.1–207.2) | 49.9 (9.1–164) | 4 |
| diabetes mellitus | 159 (49.2) | 19 (51.4) | 1 | 28 (35) | 7 (43.8) | 1 |
| antibiotic use within 1 week | 18 (5.6) | 2 (5.4) | 13 | 9 (11.3) | 3 (18.8) | 1 |
| indwelling venous catheters | 27 (8.4) | 8 (21.6) | 2 | 19 (23.8) | 10 (62.5) | 1 |
| clinically suspected bacterial endocarditis | 28 (8.7) | 8 (21.6) | 2 | 3 (3.8) | 4 (25) | 1 |
| symptoms | | | | | | |
| shaking chills | 76 (23.5) | 15 (40.5) | 2 | 14 (17.5) | 5 (31.3) | 1 |
| vomiting | 45 (13.9) | 3 (8.1) | 2 | 7 (8.8) | 2 (12.5) | 1 |

IQR: interquartile range

age (y), mean arterial pressure (mmHg), heart rate, body temperature (˚C), presence of diabetes mellitus, HD vintage (months), and white blood cell count ($\times 10^9$/L). These items were selected by clinicians as those typically used when evaluating patients for bacteremia in their daily medical practice. In validation cohort 2, it was clear at the planning stage that the respiratory rate was often missing, so the respiratory rate was excluded from the basic model.

**Added value of CPRs.** The discriminatory abilities of the basic model (Model 1), the model that added BAC-HD score (Model 2), or the model that added Shapiro's score (Model 3) to the basic model were assessed by calculating the area under the receiver operating characteristic curve (AUC) using a logistic regression model. Calibration of each model was performed based on the slope and intercept of the calibration plot [28, 29].

The number of patients correctly reclassified by adding the CPR to the basic model is shown using Net Reclassification Improvement (NRI) (Table 2). The prediction probabilities in the three models were stratified based on the tertile of the prediction probabilities of the basic model, as low ($< 0.08$), intermediate ($0.08–0.2$), or high ($> 0.2$).

**Assessment of performance.** To evaluate potential cut-off scores, we computed the sensitivity, specificity, likelihood ratio, positive predictive value, and negative predictive value for the CPRs. For brevity, only the values in validation cohort 1 are summarized.

**Handling of missing values.** All missing values were addressed using multiple imputations by chained equation treated as missing at random; ten imputed datasets were created. Three logistic regression models were conducted on each of 10 datasets and combined with Rubin's rule.

All statistical analyses were performed using Stata version 15.0 (Stata Corp., College Station, TX, USA).

## Results

### Study participants

As the final analytic cohort, there were 360 participants and 96 participants in validation cohort 1 and validation cohort 2, respectively, as shown in Fig 1.

### Added diagnostic value of BAC-HD score and Shapiro's score

In validation cohort 1, compared to the AUC of 0.69 (95% confidence interval [95% CI]: 0.59 −0.80) of Model 1 (Basic model), the AUCs of Model 2 (Basic model + BAC-HD score) and

**Table 2. Reclassification table comparing the probability of bacteremia predicted by Model 2 (basic model + BAC-HD score) and Model 3 (basic model + Shapiro's score).**

| Validation cohort 1 | Model 2: basic model[a] + BAC-HD score | | | | Model 3: basic model[a] + Shapiro's score | | | |
|---|---|---|---|---|---|---|---|---|
| Model 1: basic model[a] | <8% | 8–20% | >20% | total | <8% | 8–20% | >20% | total |
| Probability group for bacteremia | | | | | | | | |
| *Participants with bacteremia* | | | | | | | | |
| < 8% | 2 | 2 | 0 | 4 | 4 | 0 | 0 | 4 |
| 8–20% | 5 | 8 | 7 | 20 | 4 | 12 | 4 | 20 |
| > 20% | 0 | 1 | 12 | 13 | 0 | 3 | 10 | 13 |
| Total | 7 | 11 | 19 | 37 | 8 | 15 | 14 | 37 |
| *Participants without bacteremia* | | | | | | | | |
| < 8% | 127 | 11 | 0 | 138 | 114 | 24 | 0 | 138 |
| 8–20% | 79 | 63 | 16 | 158 | 55 | 89 | 14 | 158 |
| > 20% | 2 | 15 | 10 | 27 | 0 | 19 | 8 | 27 |
| total | 208 | 89 | 26 | 323 | 169 | 132 | 22 | 323 |
| Validation cohort2 | Model2: basic model[a] + BAC-HD score | | | | Model3: basic model[a] + Shapiro's score | | | |
| Model1: basic model[a] | < 8% | 8–20% | > 20% | total | < 8% | 8–20% | > 20% | total |
| Probability group for bacteremia | | | | | | | | |
| *Participants with bacteremia* | | | | | | | | |
| < 8% | 1 | 0 | 0 | 1 | 0 | 1 | 0 | 1 |
| 8–20% | 1 | 1 | 2 | 4 | 0 | 4 | 0 | 4 |
| > 20% | 0 | 1 | 10 | 11 | 0 | 2 | 9 | 11 |
| total | 2 | 2 | 12 | 16 | 0 | 7 | 9 | 16 |
| *Participants without bacteremia* | | | | | | | | |
| < 8% | 30 | 0 | 2 | 32 | 29 | 0 | 3 | 32 |
| 8–20% | 17 | 7 | 3 | 27 | 10 | 17 | 0 | 27 |
| > 20% | 0 | 9 | 12 | 21 | 4 | 5 | 12 | 21 |
| total | 47 | 16 | 17 | 80 | 43 | 22 | 15 | 80 |

Validation cohort 1:net reclassification improvement (NRI) [Model 2] 0.24−0.16+0.30−0.08 = 0.3, NRI [Model 3] 0.11− 0.16 + 0.23 − 0.12 = 0.06

Validation cohort 2: NRI [Model2] 0.13 − 0.13 + 0.33 − 0.06 = −0.27, NRI [Model 3] 0.06 − 0.13 + 0.24 − 0.04 = −0.13

basic model: logistic regression model with explanatory variables of sex (0: female, 1: male), age (y), mean arterial pressure (mmHg), pulse rate (per minute), body temperature (°C), presence of diabetes mellitus, HD vintage, white blood cell count ($\times 10^9$/L).

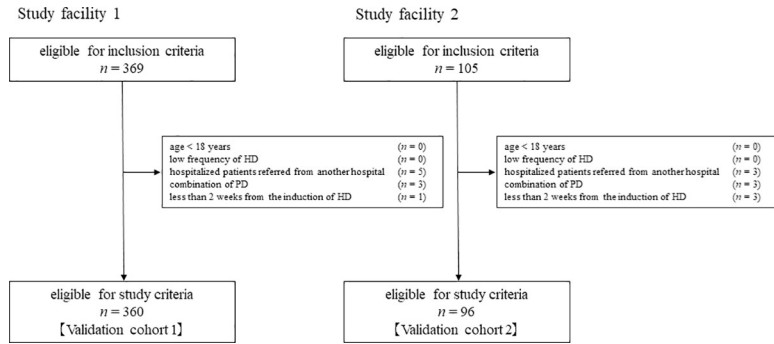

**Fig 1. Study flow.**

Model 3 (Basic model + Shapiro's score) increased to 0.8 (95% CI: 0.71−0.88) and 0.73 (95% CI: 0.63−0.83), respectively, as shown in Fig 2. In validation cohort 2, compared to the AUC of 0.81 (95% confidence interval [95% CI]: 0.68−0.94) of Model 1 (Basic model), the AUCs of Model 2 (Basic model + BAC-HD score) and Model 3 (Basic model + Shapiro's score) increased to 0.83 (95% CI: 0.72−0.95) and 0.85 (95% CI: 0.76−0.94), respectively, as shown in Fig 2.

In validation cohort 1, the slopes (intercept) of the calibration plots of Models 1, 2, and 3 were 1.19 (−0.20), 1.09 (−0.01) and 1.15 (−0.01), respectively. In validation cohort 2, the slopes (intercept) of the calibration plots of Models 1, 2 and 3 were 1.03 (−0.01), 1.00 (−0.001) and 0.86 (0.03), respectively. Calibration plots are shown in Fig 3.

## Reclassification by BAC-HD score and Shapiro's score

In validation cohort 1 (N = 360), 138 of 142 patients predicted to have a low probability of bacteremia as a result of the basic model (Model 1) were not bacteremic (negative predictive value [NPV] = 97.1%). Thirteen of the 40 patients predicted to have a high probability of bacteremia had bacteremia (positive predictive value; PPV 32.5). Adding the BAC-HD score (Model 2) increased the number of patients predicted to be low-probability from 142 to 215, of whom 208 were not bacteremic (NPV 96.7%). The number of patients predicted to have a high probability of bacteremia was increased from 40 to 45, of whom 19 were bacteremic (PPV 42.2%). Adding Shapiro's score (Model 3) increased the number of patients predicted to be low-probability from 142 to 177, of whom 169 were not bacteremic (NPV 95.4%). The number of patients predicted to be high probability was reduced from 40 to 36, of whom 14 were

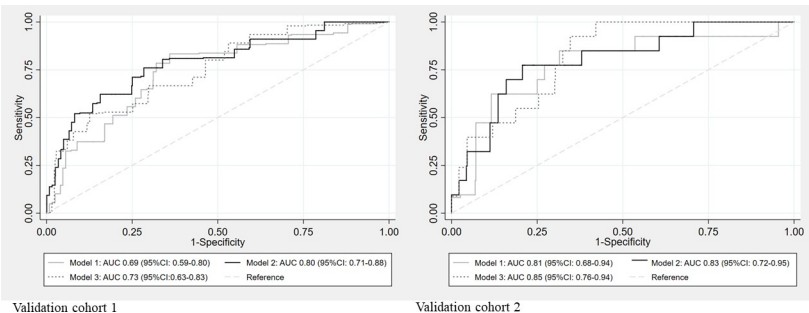

**Fig 2. Receiver operating characteristic curve.** A: results for validation cohort 1, B: results for validation cohort 2. Model 1 = basic model, Model 2 = basic model + BAC-HD score, Model 3 = basic model + Shapiro's score, AUC: area under the receiver operating characteristic curve, 95% CI: 95% confidence interval.

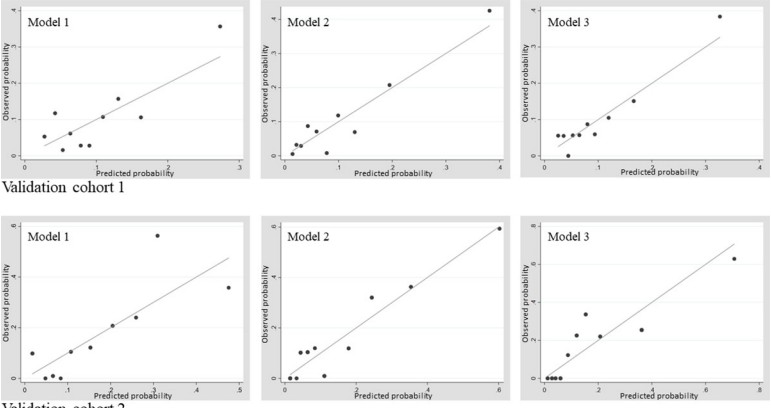

**Fig 3. Calibration plot.** Model 1 = basic model, Model 2 = basic model + BAC-HD score, Model 3 = basic model + Shapiro's score.

bacteremic (PPV 38.9%). The NRI values for addition of BAC-HD score and Shapiro's score were 0.3 and 0.06, respectively, in validation cohort 1. The results for validation cohort 2 are shown in Table 2 and the NRIs for addition of BAC-HD score and Shapiro's score were 0.27 and 0.13, respectively.

## Assessment of performance

The sensitivities, specificities, likelihood ratios and predictive values for possible cut-off scores in CPRs in validation cohort 1 are shown in Table 3.

## Discussion

This is the first study to validate and compare the external validity of the prediction rules for bacteremia in HD patients. BAC-HD showed an excellent added value, suggesting that it can

**Table 3. Assessment of performance of BAC-HD and Shapiro's score in validation cohort 1.**

| BAC–HD score | | | | | | | | | | | | |
|---|---|---|---|---|---|---|---|---|---|---|---|---|
| Cut–off | Sensitivity | 95%CI | Specificity | 95%CI | LR+ | 95%CI | LR− | 95%CI | PPV | 95%CI | NPV | 95%CI |
| ≥ 4 | 5.2 | (3.2–7.8) | 94.6 | (93.8–95.3) | 1 | (0.6–1.5) | 1 | (1.0–1.0) | 9.8 | (6.2–14.6) | 89.7 | (88.7–90.7) |
| ≥ 3 | 21.1 | (17.3–25.4) | 92.5 | (91.5–93.3) | 2.8 | (2.3–3.5) | 0.9 | (0.8–0.9) | 24.3 | (19.9–29.1) | 91.1 | (90.1–92) |
| ≥ 2 | 63.6 | (58.8–68.3) | 78.2 | (76.8–79.6) | 2.9 | (2.7–3.2) | 0.5 | (0.4–0.5) | 25.1 | (22.5–27.8) | 94.9 | (94.1–95.7) |
| ≥ 1 | 85.7 | (82–89) | 35.2 | (33.6–36.8) | 1.3 | (1.3–1.4) | 0.4 | (0.3–0.5) | 13.2 | (11.9–14.5) | 95.6 | (94.3–96.6) |
| Shapiro's score | | | | | | | | | | | | |
| Cut–off | Sensitivity | 95%CI | Specificity | 95%CI | LR+ | 95%CI | LR− | 95%CI | PPV | 95%CI | NPV | 95%CI |
| ≥ 9 | 5.7 | (3.6–8.4) | 99.2 | (98.8–99.5) | 6.9 | (4.0–11.9) | 1 | (0.9–1.0) | 44.2 | (30.5–58.7) | 90.2 | (89.2–91.1) |
| ≥ 8 | 11.3 | (8.4–14.8) | 97 | (96.4–97.6) | 3.8 | (2.7–5.3) | 0.9 | (0.9–1.0) | 30.3 | (23.1–38.2) | 90.5 | (89.5–91.4) |
| ≥ 7 | 24.8 | (20.7–29.3) | 94.7 | (93.9–95.4) | 4.6 | (3.7–5.8) | 0.8 | (0.8–0.8) | 34.7 | (29.2–40.5) | 91.7 | (90.7–92.5) |
| ≥ 6 | 39.6 | (34.8–44.5) | 90.9 | (89.9–91.9) | 4.4 | (3.7–5.1) | 0.7 | (0.6–0.7) | 33.3 | (29.1–37.7) | 92.9 | (92.0–93.8) |
| ≥ 5 | 45.9 | (41.0–50.9) | 85 | (83.8–86.1) | 3.1 | (2.7–3.5) | 0.6 | (0.6–0.7) | 25.9 | (22.8–29.3) | 93.2 | (92.3–94.1) |
| ≥ 4 | 54.1 | (49.1–59.0) | 71.4 | (69.9–72.9) | 1.9 | (1.7–2.1) | 0.6 | (0.6–0.7) | 17.8 | (15.7–20.0) | 93.1 | (92.1–94.1) |
| ≥ 3 | 86.5 | (82.8–89.7) | 41.6 | (40–43.2) | 1.5 | (1.4–1.6) | 0.3 | (0.3–0.4) | 14.5 | (13.1–16.0) | 96.4 | (95.4–97.3) |
| ≥ 2 | 89.2 | (85.8–92) | 11.7 | (10.6–12.8) | 1 | (1.0–1.0) | 0.9 | (0.7–1.2) | 10.4 | (9.4–11.4) | 90.4 | (87.3–92.9) |

95%CI: 95% confidence interval, LR+: positive likelihood ration, LR−: negative likelihood ratio, PPV: positive predictive value, NPV: negative predictive value

be a useful tool for improving the diagnostic ability of bacteremia in dialysis patients. In addition, BAC-HD is a simple CPR consisting of five items, and has the advantage of being highly versatile in clinical settings.

On the other hand, although Shapiro's score has many components, its added value was inferior to the BAC-HD score. The reason for this was considered to be the effects of three characteristics of patients on maintenance HD. The first is the difference in the bacteremia etiology. Bacteremia in the general population is often due to gram-negative rods (GNR) [19–22], usually due to urinary tract infections (UTI) [1, 30], while patients on HD often have gram-positive cocci (GPC) bacteremia due to cutaneous infections [23–25]. In our experience, UTI and BSI often have different clinical presentations. Therefore, it is considered that Shapiro's score, a CPR developed in a population with a high rate of UTI, could not predict bacteremia in maintenance HD patients with a high rate of cutaneous infections.

This is consistent with our previous studies showing that the systemic inflammatory response syndrome (SIRS) criteria [31] and the quick Sequential (Sepsis-Related) Organ Failure Assessment (qSOFA) score [32, 33] were not useful in predicting bacteremia in HD patients. Second, the clinical information of patients on maintenance HD such as body weight, vital signs [34], electrolytes, blood urea nitrogen, and creatinine vary greatly between dialysis and non-dialysis days. Since bacteremia is assumed to develop regardless of the timing of dialysis, the predictive ability of these values may be impaired. Third, patients with end-stage renal failure are often immunocompromised and may have different clinical presentations [35].

This study has two strengths. First, since we verified the added value in two validation cohorts, the robustness of the results is likely increased. Second, the basic model was used to show the added value of CPRs. Although some external validation studies only showed discrimination and calibration of the CPR itself, this is not enough to evaluate the degree of improvement in predictive ability [36].

This study also has some limitations. First, since only Japanese patients with HD were included, its validity in other ethnic groups is unknown. On the other hand, AV-fistula (AVF) is used as vascular access in more than 93% [37] of Japanese on HD, satisfying the 65% goal of the fistula first initiative [38]. Since AVF is associated with a lower risk of infection compared to other vascular access methods, especially central venous catheters [39], it seems significant that added value was shown in a population with a high proportion of AVFs, which is the desired result. Second, since this study analyzed retrospective medical data, there are risks of bias caused by missing values or lower measurement accuracy. However, we performed multiple imputations for missing data to minimize such a bias [40]. Prospective studies are needed for better verification. Third, since bands, which were one of the components of Shapiro's score, were not evaluated, the score may have been underestimated. Since a number of facilities cannot always measure the proportion of bands, such as facilities included in a previous validation study of Shapiro's score [12], this is considered acceptable in terms of versatility. Since there are studies (including this present study) that have modified the items included in Shapiro's score, future validation of these is awaited [41]. Fourth, since there were no standard criteria for when to obtain blood cultures, the possibility that there was some degree of arbitrariness in the decision to draw blood cultures cannot be denied. However, since we used data before the development of the BAC-HD score, it is unlikely that the items in the BAC-HD score influenced the decision on whether or not to draw blood cultures. Furthermore, it is possible that some of the patients whose blood cultures were not drawn included cases of bacteremia. Fifth, it is unclear whether blood cultures were collected from a central venous catheter (CVC) that was the site of vascular access. However, since it is unlikely that both sets of blood cultures were collected from the CVC, and the number of CVCs was small, we believe that the effect of this was not significant.

## Conclusions

We verified the added value of the BAC-HD score and Shapiro's score to the usual criteria for predicting bacteremia in HD patients. We suggest that either the BAC-HD score or Shapiro's score may improve the accuracy of predicting bacteremia in patients on HD. Reclassification was better with the BAC-HD score.

Improving the diagnostic ability is expected to contribute to the early initiation of appropriate treatment and improve the prognosis of bacteremia.

## Supporting information

**S1 Table. Pathogens involved in bacteremia in validation cohorts 1 and 2.**
(DOCX)

**S1 Data. Validation cohort 1.**
(XLS)

**S2 Data. Validation cohort 2.**
(XLS)

## Acknowledgments

The authors thank the JOINT-KD collaborators, Ryo Nishioka and Yasunori Suzuki, Department of Rheumatology, Kanazawa University Hospital, Kanazawa and Naomi Ako, Division of Nephrology and Hypertension, Department of Internal Medicine, St. Marianna University School of Medicine, Kawasaki for their intellectual support in the management of this study. We also thank Libby Cone, MD, MA, of DMC Corp. (www.dmed.co.jp) for editing drafts of this manuscript.

## Author Contributions

**Conceptualization:** Sho Sasaki.

**Data curation:** Sho Sasaki, Yoshihiko Raita, Minoru Murakami, Shungo Yamamoto, Kentaro Tochitani.

**Formal analysis:** Sho Sasaki.

**Methodology:** Sho Sasaki, Yoshihiko Raita, Shungo Yamamoto, Kentaro Tochitani, Takeshi Hasegawa.

**Software:** Sho Sasaki.

**Supervision:** Yoshihiko Raita, Minoru Murakami, Takeshi Hasegawa, Kiichiro Fujisaki, Shunichi Fukuhara.

**Validation:** Sho Sasaki.

**Writing – original draft:** Sho Sasaki, Yoshihiko Raita, Shungo Yamamoto, Kentaro Tochitani.

**Writing – review & editing:** Sho Sasaki, Yoshihiko Raita, Takeshi Hasegawa, Kiichiro Fujisaki, Shunichi Fukuhara.

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
