## [Decision Letter · Decision Letter 0]

9 Dec 2020

PONE-D-20-33423

Added value of clinical prediction rules for bacteremia in hemodialysis patients: An external validation study

PLOS ONE

Dear Dr. Sasaki,

Thank you for submitting your manuscript to PLOS ONE. After careful consideration, we feel that it has merit but does not fully meet PLOS ONE’s publication criteria as it currently stands. Therefore, we invite you to submit a revised version of the manuscript that addresses the points raised during the review process.

The topic is potentially interesting and important issue.  The three experts raised some concerns and points to be clearly described.

We look forward to receiving your revised manuscript.

Kind regards,

Tatsuo Shimosawa, M.D., Ph.D.

Academic Editor

PLOS ONE

Journal Requirements:

2. Thank you for stating in the text of your manuscript The present study was approved by the ethics committee of Iizuka Hospital (17167-436), Okinawa Prefectural Chubu Hospital (H28-51) and Saku General Hospital (R201701-01). The study was conducted in accordance with the ethical standards of the Declaration of Helsinki." Please also add this information to your ethics statement in the online submission form.

3. Please confirm whether ethical approval was obtained for the original cohort and for the validation cohorts, or whether there was a different ethical approval for cohorts 1&2.

4. Thank you for stating in your manuscript "Since all patient information analyzed in this study was retrospective, participants’ written informed consent was not obtained." In your ethics statement in the Methods section and in the online submission form, please clarify whether all data were fully anonymized before or after you accessed them and/or whether the IRB or ethics committee waived the requirement for informed consent.

5. Please include the date(s) on which you accessed the databases or records to obtain the data used in your study.

6. We note that you have indicated that data from this study are available upon request. PLOS only allows data to be available upon request if there are legal or ethical restrictions on sharing data publicly. For information on unacceptable data access restrictions, please see http://journals.plos.org/plosone/s/data-availability#loc-unacceptable-data-access-restrictions.

Reviewers' comments:

Reviewer's Responses to Questions

**Comments to the Author**

1. Is the manuscript technically sound, and do the data support the conclusions?

Reviewer #1: Yes

Reviewer #2: Yes

Reviewer #3: Yes

2. Has the statistical analysis been performed appropriately and rigorously? 

Reviewer #1: Yes

Reviewer #2: Yes

Reviewer #3: Yes

3. Have the authors made all data underlying the findings in their manuscript fully available?

Reviewer #1: No

Reviewer #2: No

Reviewer #3: Yes

4. Is the manuscript presented in an intelligible fashion and written in standard English?

Reviewer #1: Yes

Reviewer #2: Yes

Reviewer #3: Yes

5. Review Comments to the Author

Reviewer #1: Dear authors,

Predicting bacteria in in HD patients is important clinical tool to care for this patient population. The methodology is standard and statistical analysis is robust. Minor comments are as follows.

1- Why was the study design as a basic model and added models? Why not the comparison between the 2 final models? Was the basic model validated to start with? (To assess the value of reclassification)

2- A typo in second line of introduction (Key should be keys).

3- How does the lack of standard criteria for blood cultures make the 2 cohorts different in characteristics?

4- How did the expert panel exclude the contamination from retrospective data?

5- Please explain why difference in etiology (Gram-negative vs Gram-positive) between genera population and HD patient will render Shapro's score ineffective in predicting bacteremia.

6- Please explain the calibration plot.

Regards

Reviewer #2: This is an interesting study addressing the external validation of a novel clinical prediction rule (CPR) for bacteremia (BAC-HD score) which the authors have previously developed specifically for hemodialysis (HD) patients. They compared the diagnostic accuracy of BAC-HD score and the Shapiro’s criteria and reported that the former was better in the discrimination ability to predict bacteremia than the latter. The article is well drafted and certainly provides valuable information to supplement clinical judgment and treatment decision in the dialysis unit. The reviewer has a few comments as follows:

1) The indications for obtaining blood cultures are not presented, and the patient selection for the two cohorts cannot escape a certain degree of arbitrariness. There may have been patients with bacteremia who did not have a blood culture obtained and therefore were not included I the patient population. This point should be clearly stated as limitation in the manuscript.

2) The authors refer to the uniqueness of bacteria etiology of HD patients. Can the authors present data of the frequency of the organisms actually cultured from the blood culture of the patients? Can the authors present data concerning the suspected infectious foci of the patients as well?

3) Concerning the ‘false negative’ cases (i.e. score did not suggest a culture but the patient was found to be bacteremic), there seems to be seven patients in Model 2 and eight in Model 2. Do these patients of the two Models show considerable overlapping? Can the authors describe the clinical circumstances of these patients and comment on the reason why the patients were missed by the prediction rule?

Reviewer #3: This study intends to provide exteral validation to a recently described prediction model for diagnosing bacteremia in hemodialysis patients. Further, the diagnostic yield of the new model is compared to an established model by Shapiro et al.

Since bacteremia is an important problem and the diagnosis is not straight forward, this approach is relevant and interesting. The paper is well written.

The study is retrospective in design, two validation cohorts were formed at two different hospitals. Patients were included if they were on maintenance hemodialysis and two blood cultures were drawn at hospital admission. The prediction models were compared with the results of the blood cultures.

Both additive models improved prediction of positive blood cultures. The authors claim that their new score performed better than Shapiro's rule, however, the data suggest that both rules add similarly to the basic model. The authors should check if they really can assume superiority.

Some suggestions to the authors:

1) line 61: fortunately bacteremia is not identical with sepsis, although it may lead to this severe complication. The mortality rates cited in #18 and #19 are for Sepsis.

2) lines 65 f: please consider rewording, this sentence is hard to understand. Further, at least with CVC the time point of dialysis is most likely irrelevant for the diagnostic yield of blood cultures when drawn from the catheter.

3) Would the addition of the condition "CVC present" further improve the BAC-HD score? This seems most likely given the numbers in Tbl. 1

6. PLOS authors have the option to publish the peer review history of their article (what does this mean?). If published, this will include your full peer review and any attached files.

Reviewer #1: No

Reviewer #2: No

Reviewer #3: No

---

## [Author Response · Author response to Decision Letter 0]

22 Jan 2021

RESPONSE TO EDITOR

RESPONSE: Thank you for your very clear guidance. We have referred to the guide and revised the manuscript accordingly.

2. Thank you for stating in the text of your manuscript The present study was approved by the ethics committee of Iizuka Hospital (17167-436), Okinawa Prefectural Chubu Hospital (H28-51) and Saku General Hospital (R201701-01). The study was conducted in accordance with the ethical standards of the Declaration of Helsinki." Please also add this information to your ethics statement in the online submission form.

RESPONSE: Thank you for reminding us of this; we will include this on the submission form,. 

3. Please confirm whether ethical approval was obtained for the original cohort and for the validation cohorts, or whether there was a different ethical approval for cohorts 1&2.

RESPONSE: The validation cohort and the original cohort were separately approved by the ethics committees of the relevant institutions. In this article, we have provided the approval numbers for the validation cohort.

4. Thank you for stating in your manuscript "Since all patient information analyzed in this study was retrospective, participants’ written informed consent was not obtained." In your ethics statement in the Methods section and in the online submission form, please clarify whether all data were fully anonymized before or after you accessed them and/or whether the IRB or ethics committee waived the requirement for informed consent. Please include the date(s) on which you accessed the databases or records to obtain the data used in your study.

RESPONSE: Thank you for your suggestion. We have revised the sentence in the Methods section. We will also include additional information on the submission form. 

Revised sentences (lines 83–87):

Since all patient information analyzed in this study was retrospective, participants’ written informed consent was not required by the ethics committee. All data were fully anonymized before authors accessed them. We accessed the medical records at Okinawa Prefectural Chubu Hospital from February 9th to February 13th, 2017, and at Saku General Hospital from February 24th to 26th, 2017. 

RESPONSE: 

We have obtained permission to provide the minimal anonymized datasets from the ethical committees of participating facilities.

REVIEWER 1:

1- Why was the study design as a basic model and added models? Why not the comparison between the 2 final models? Was the basic model validated to start with? (To assess the value of reclassification)

REPONSE: We wish to express our appreciation to Reviewer 1 for his or her insightful comments, which have helped us significantly improve the paper.

We appreciate your raising this important point. As you pointed out, we created a basic model to assess the added value of clinical prediction rules (CPRs). We referred to previous studies (Moons, 2012 #12226), (Takada, 2020 #12228) as a basis for considering that validation of the basic model is not necessary. We have revised line 141 as below:

Revised sentence (lines 142–145):

The basic model to assess the value of reclassification of CPRs was conducted using a logistic regression model with explanatory variables of sex (0: female, 1: male), age (y), mean arterial pressure (mmHg), heart rate, body temperature (°C), presence of diabetes mellitus, HD vintage (months), and white blood cell count (× 109/L).

2- A typo in second line of introduction (Key should be keys).

RESPONSE:

We apologize for our mistake and have corrected it (Line 55). 

3- How does the lack of standard criteria for blood cultures make the 2 cohorts different in characteristics?

RESPONSE:

We appreciate your raising this important point. Unfortunately, it is difficult to provide data on the standard procedures for the drawing of blood cultures at different facilities. Clinically, the threshold for blood cultures being drawn varies among institutions. We have modified line 123 as follows:

Revised sentence (line 128):

Since there are no standard criteria for obtaining blood cultures, it was suspected that the selection of subjects may have differed depending on the facility.

4- How did the expert panel exclude the contamination from retrospective data?

RESPONSE:

We appreciate your important point. We apologize for the lack of explanation in the text. The expert panel, which was blinded to the study design, used the results of the blood cultures, as well as basic patient information and blood test results, to determine contamination. We have revised lines 117–121 accordingly:

Revised sentences (lines 119–123):

Finally, an external consensus panel of infectious disease physicians with > 10 y clinical experience and Japanese Board of Infectious Disease certification who were blinded to the present study design determined whether a culture was contaminated or not based on the above definitions and their clinical expertise.

5- Please explain why difference in etiology (Gram-negative vs Gram-positive) between genera population and HD patient will render Shapiro's score ineffective in predicting bacteremia.

RESPONSE:

We appreciate Reviewer 1’s pointing out this important issue. Unfortunately, we have not found adequate evidence that different species or foci of infection have different clinical presentations. We have revised accordingly:

Revised text (lines 227–230):

In our experience, UTIs and BSIs often have different clinical presentations. We therefore considered that Shapiro’s score, a CPR developed in a population with a high rate of UTIs, would be unable to predict bacteremia in maintenance HD patients with a high rate of cutaneous infections.

6- Please explain the calibration plot.

We apologize to reviewer 1 for the lack of explanation. The calibration plot is a diagram for evaluating the calibration of a logistic regression model, checking the difference between the observed probability and the probability estimated by the regression model. Therefore, the closer the slope of the plot is to 1 and the closer the intercept is to 0, the better the calibration is.

REVIEWER 2:

1) The indications for obtaining blood cultures are not presented, and the patient selection for the two cohorts cannot escape a certain degree of arbitrariness. There may have been patients with bacteremia who did not have a blood culture obtained and therefore were not included I the patient population. This point should be clearly stated as limitation in the manuscript.

We wish to express our appreciation to Reviewer 2 for his or her insightful comments, which have helped us significantly improve the paper.

RESPONSE TO REVIEWER 2:

We appreciate your important point. We added the following to the limitations (line 261–267) as suggested by reviewer 2:

Fourth, since there were no standard criteria for when to obtain blood cultures, the possibility that there was some degree of arbitrariness to the decision to draw blood cultures cannot be denied. However, since we used data before the development of the BAC-HD score, the items in the BAC-HD score could not have influenced the decision on whether or not to draw blood cultures. Furthermore, it is possible that some of the patients whose blood cultures were not drawn included cases of bacteremia.

2) The authors refer to the uniqueness of bacteria etiology of HD patients. Can the authors present data of the frequency of the organisms actually cultured from the blood culture of the patients? Can the authors present data concerning the suspected infectious foci of the patients as well?

RESPONSE:

We thank reviewer 2 for these important points. We have described the frequency of the causative organism in S1 Table. Unfortunately, we do not have reliable data on suspected infectious foci due to the fact that this is a retrospective study, so we are unable to provide data on this.

New table: 

S1 Table. Pathogens involved in bacteremia in validation cohorts 1 and 2

 Validation cohort 1

n = 37 Validation cohort 2

n = 16 Total

n = 53

Staphylococcus aureus (S. aureus) 9 (24.4) 9 (56.2) 18 (34.0)

[methicillin-resistant S. aureus] 5 2 7

coagulase negative Staphylococci 1 (2.7) 2 (12.5) 3 (5.7)

Streptococcus spp. 2 (5.4) 2 (12.5) 4 (7.5)

Enterococcus spp. 2 (5.4) 2 (12.5) 4 (7.5)

Escherichia coli 11 (29.7) 0 11 (20.8)

Klebsiella pneumoniae 4 (10.8) 1 (6.3) 5 (9.4)

Others 8 (21.6) 0 8 (15.1)

3) Concerning the ‘false negative’ cases (i.e. score did not suggest a culture but the patient was found to be bacteremic), there seems to be seven patients in Model 2 and eight in Model 2. Do these patients of the two Models show considerable overlapping? Can the authors describe the clinical circumstances of these patients and comment on the reason why the patients were missed by the prediction rule?

RESPONSE:

We thank Reviewer 2 for these important questions. Two of the false-negative cases overlapped. Unfortunately, we could not find any similarities between these cases, and the reason why the predictive scores showed false negatives is unknown.

REVIEWER 3:

 This study intends to provide external validation to a recently described prediction model for diagnosing bacteremia in hemodialysis patients. Further, the diagnostic yield of the new model is compared to an established model by Shapiro et al.

Since bacteremia is an important problem and the diagnosis is not straight forward, this approach is relevant and interesting. The paper is well written.

The study is retrospective in design, two validation cohorts were formed at two different hospitals. Patients were included if they were on maintenance hemodialysis and two blood cultures were drawn at hospital admission. The prediction models were compared with the results of the blood cultures.

Both additive models improved prediction of positive blood cultures. The authors claim that their new score performed better than Shapiro's rule, however, the data suggest that both rules add similarly to the basic model. The authors should check if they really can assume superiority.

We wish to express our appreciation to Reviewer 3 for his or her insightful comments, which have helped us significantly improve the paper.

We appreciate your important point. As you noted, the interpretation of the results was incorrect. The following corrections have been made:

Revised sentence (lines 47–49):

Either the BAC-HD score or Shapiro’s score may improve the ability to diagnose bacteremia in HD patients. Reclassification was better with the BAC-HD score.

Revised sentence (lines 274–276): 

We suggest that either the BAC-HD score or Shapiro’s score may improve the accuracy of predicting bacteremia in patients on HD. Reclassification was better with the BAC-HD score.

Some suggestions to the authors:

1) line 61: fortunately bacteremia is not identical with sepsis, although it may lead to this severe complication. The mortality rates cited in #18 and #19 are for Sepsis.

RESPONSE:

We appreciate the important point made by reviewer 3. We apologize for our inaccurate description. We have revised line 65 as follows:

The annual mortality due to sepsis, a severe complication of bacteremia, in HD patients is 50–100 times higher than that of the general population

2) lines 65 f: please consider rewording, this sentence is hard to understand. Further, at least with CVC the time point of dialysis is most likely irrelevant for the diagnostic yield of blood cultures when drawn from the catheter.

RESPONSE: 

We appreciate your important point. We have revised lines 69–72. In addition, we added sentences to the limitations (line 257) that blood cultures may have been taken from the CVC. However, in our opinion, since only a small number of Japanese dialysis patients use CVC, the impact of CVC on the present study was small.

Revised sentence (lines 69–72):

Since many blood test findings in HD patients are often affected by dialysis, the accuracy of items included in existing CPRs developed in the general population, such as blood counts and serum creatinine levels, may be greatly affected

New sentence (lines 267–271):

Fifth, it is unclear whether blood cultures were collected from a central venous catheter (CVC) that was the site of vascular access. However, since it is unlikely that both sets of blood cultures were collected from the CVC, and the number of CVCs was small, we believe that the effect of this was not significant.

3) Would the addition of the condition "CVC present" further improve the BAC-HD score? This seems most likely given the numbers in Tbl. 1

RESPONSE: 

Thank you for pointing out this very important issue. In the BAC-HD development paper, we used “CVC use” as a candidate predictor, but it did not remain in the final model. Since the purpose of this study was to test external validity, we did not test the effect of CVC use on improving predictive ability. There are two reasons why CVC use was not included in the final model even though it is an important predictor, the first being that it was incompatible with other variables in multivariate models, and the second that the number of CVCs used was too small.

---

## [Decision Letter · Decision Letter 1]

10 Feb 2021

Added value of clinical prediction rules for bacteremia in hemodialysis patients: An external validation study

PONE-D-20-33423R1

Dear Dr. Sasaki,

We’re pleased to inform you that your manuscript has been judged scientifically suitable for publication and will be formally accepted for publication once it meets all outstanding technical requirements.

Kind regards,

Tatsuo Shimosawa, M.D., Ph.D.

Academic Editor

PLOS ONE

Additional Editor Comments (optional):

Reviewers' comments:

Reviewer's Responses to Questions

**Comments to the Author**

1. If the authors have adequately addressed your comments raised in a previous round of review and you feel that this manuscript is now acceptable for publication, you may indicate that here to bypass the “Comments to the Author” section, enter your conflict of interest statement in the “Confidential to Editor” section, and submit your "Accept" recommendation.

Reviewer #1: All comments have been addressed

Reviewer #2: All comments have been addressed

2. Is the manuscript technically sound, and do the data support the conclusions?

Reviewer #1: Yes

Reviewer #2: Yes

3. Has the statistical analysis been performed appropriately and rigorously? 

Reviewer #1: Yes

Reviewer #2: Yes

4. Have the authors made all data underlying the findings in their manuscript fully available?

Reviewer #1: Yes

Reviewer #2: Yes

5. Is the manuscript presented in an intelligible fashion and written in standard English?

Reviewer #1: Yes

Reviewer #2: Yes

6. Review Comments to the Author

Reviewer #1: Dear authors,

Thank you for addressing my previous comments. I do not have any other comments at this point.

Best,

Reviewer #2: (No Response)

7. PLOS authors have the option to publish the peer review history of their article (what does this mean?). If published, this will include your full peer review and any attached files.

Reviewer #1: **Yes: **Islam M. Ghazi, PharmD

Reviewer #2: No

---

## [Editor Report · Acceptance letter]

12 Feb 2021

PONE-D-20-33423R1 

Added value of clinical prediction rules for bacteremia in hemodialysis patients: An external validation study 

Dear Dr. Sasaki:

I'm pleased to inform you that your manuscript has been deemed suitable for publication in PLOS ONE. Congratulations! Your manuscript is now with our production department. 

Kind regards, 

on behalf of

Prof. Tatsuo Shimosawa 

Academic Editor

PLOS ONE